# Magnetic field-tuned Fermi liquid in a Kondo insulator

Satya K. Kushwaha [1,2], Mun K. Chan [1], Joonbum Park[1], S.M. Thomas [2], Eric D. Bauer [2], J.D. Thompson[2], F. Ronning[2], Priscila F.S. Rosa[2] & Neil Harrison[1]*

Kondo insulators are expected to transform into metals under a sufficiently strong magnetic field. The closure of the insulating gap stems from the coupling of a magnetic field to the electron spin, yet the required strength of the magnetic field–typically of order 100 T–means that very little is known about this insulator-metal transition. Here we show that $Ce_3Bi_4Pd_3$, owing to its fortuitously small gap, provides an ideal Kondo insulator for this investigation. A metallic Fermi liquid state is established above a critical magnetic field of only $B_c \approx 11$ T. A peak in the strength of electronic correlations near $B_c$, which is evident in transport and susceptibility measurements, suggests that $Ce_3Bi_4Pd_3$ may exhibit quantum criticality analogous to that reported in Kondo insulators under pressure. Metamagnetism and the breakdown of the Kondo coupling are also discussed.

[1] MPA-MAGLAB, Los Alamos National Laboratory, Los Alamos, NM 87545, USA. [2] MPA-CMMS, Los Alamos National Laboratory, Los Alamos, NM 87545, USA. *email: nharrison@lanl.gov

**K**ondo insulators are a class of quantum materials in which the coupling between conduction electrons and nearly localized $f$-electrons may lead to properties that are distinct from those of conventional band insulators[1–3]. Remarkable properties of current interest include topologically protected surface states that are predicted[4–6] and reportedly confirmed by experiments[7–12], and reports of magnetic quantum oscillations originating from the insulating bulk[13–16]. The very same magnetic field that produces quantum oscillations also couples to the $f$-electron magnetic moments, driving the Kondo insulator inexorably towards a metallic state[17,18]. The required magnetic fields in excess of 100 T[17,19] have, however, proven to be prohibitive for a complete characterization of the metallic ground state.

Thermodynamic experiments have thus far provided evidence for the presence of electronic correlations in Kondo insulators at high-magnetic fields. Quantum oscillation experiments, for example, have found moderately heavy masses within the insulating phase in strong magnetic fields[15]. Furthermore, heat capacity experiments have shown that the electronic contribution undergoes an abrupt increase with increasing magnetic field[20,21]: in one case[20] the increase occurs within the insulating phase, suggesting the presence of in-gap states[22], whereas in another, it coincides[21] with the onset of an upturn in the magnetic susceptibility[18,19] and reports of metallic behavior[7,18]. An unambiguous signature of metallic behavior, such as electrical resistivity that increases with increasing temperature at accessible magnetic fields, has yet to be reported.

We show here that insulating $Ce_3Bi_4Pd_3$, once polished to remove surface contamination, exhibits properties consistent with it being a reduced gap variant of its sister Kondo insulator $Ce_3Bi_4Pt_3$[22,23], which is also a topological Kondo insulator candidate[4,24,25]. The atypically small magnetic field of $B_c \approx 11$ T required to overcome the Kondo gap of insulating $Ce_3Bi_4Pd_3$ (see Fig. 1), makes this material[26] ideal for investigating the metallization of a Kondo insulator in strong magnetic fields. We find a magnetic field-induced metallic state exhibiting an electrical resistivity that varies as $T^2$ at low temperatures (where $T$ is the temperature), thereby revealing a Fermi liquid ground state in the high-magnetic field metallic phase. We also identify a magnetic field-tuned collapse of the Fermi liquid temperature scale $T_{FL}$ near $B_c$, indicating that $Ce_3Bi_4Pd_3$ may exhibit a magnetic field-tuned quantum critical point analogous to that observed in $SmB_6$ as a function of pressure[27–31]. The origin of the collapsing Fermi liquid temperature scale in $Ce_3Bi_4Pd_3$ is revealed by susceptibility measurements, which find a peak as a function of magnetic field at $B_c$ that can be traced back to a broad maximum in the susceptibility as a function of temperature at $T_M$ in weak magnetic fields. This maximum implies the formation of Kondo singlets at a temperature $T_K = cT_M$ (where $3 < c < 4$)[20], yielding $15 < T_K < 20$ K. A gradual evolution of the susceptibility from a crossover along the temperature axis to a sharper peak along the magnetic field axis at $B_c$ is therefore unveiled.

## Results

**Evidence for a small gap Kondo insulator.** Typical signatures of a Kondo insulating state include an electrical resistivity that exhibits a thermally activated insulating $\partial \rho_{xx}/\partial T < 0$ behavior (where $\rho_{xx}$ is the electrical resistivity) over a wide range of temperatures and a Curie–Weiss behavior in the magnetic susceptibility at high temperatures that crosses over via a maximum in the susceptibility into a Kondo gapped state at low temperatures[1,3,22,32–35]. Zero magnetic field electrical resistivity and magnetic susceptibility measurements reveal $Ce_3Bi_4Pd_3$ to exhibit such signatures (see Figs. 1 and 2 and Supplementary Fig. 1), but with a Kondo temperature scale that is significantly reduced relative to those found in the archetypal Kondo insulators $SmB_6$ and $Ce_3Bi_4Pt_3$. Typical

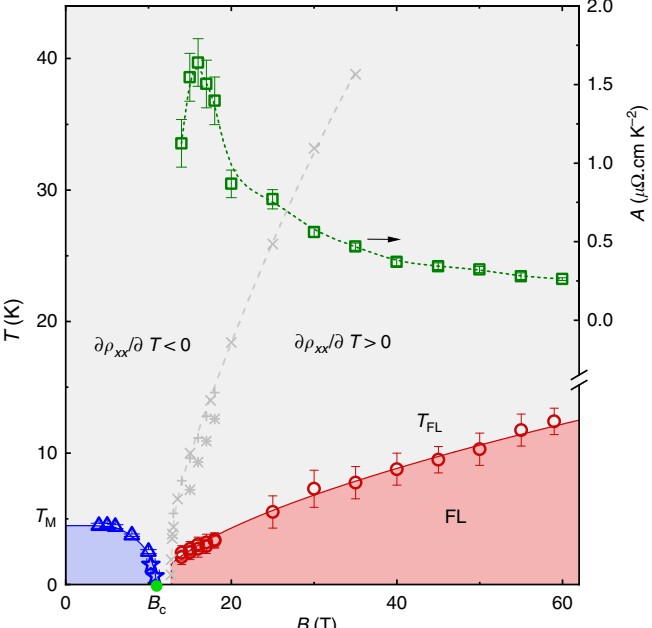

**Fig. 1** Magnetic field versus temperature phase diagram of $Ce_3Bi_4Pd_3$. $T_M$ indicates the peak in the susceptibility ($\chi$) obtained from $\chi$ versus $T$ (triangles) and $B$ (stars) from Fig. 6. Here, $B_c = B(T_M \to 0)$ is indicated by a green circle. Also shown is the temperature $T_{FL}$ (circles) above which the resistivity departs from $\rho_{xx} = \rho_0 + AT^2$ (i.e., Fermi liquid) behavior. Shaded regions and lines are drawn as guides to the eye. The gray dashed line shows where the derivative $\partial \rho_{xx}/\partial T = 0$ (plus symbols from sample A in static fields, × symbols from sample A in pulsed fields and asterisk symbols from sample B in static fields; see Methods). To the left of this line $\partial \rho_{xx}/\partial T < 0$, indicative of insulating behavior, while to the right of this line $\partial \rho_{xx}/\partial T > 0$, indicative of metallic behavior. The right-hand-axis indicates the magnetic field-dependence of the $A$ coefficient (squares), with the values for $B \lesssim 20$ T corresponding to the average of samples A and B in Fig. 5. Error bars represent the standard error.

Kondo insulators have Kondo gaps of several millielectronvolts, accompanied by inverse residual resistivity ratios in the range $10^2 < R_{RRR}^{-1} < 10^5$ and maxima in the magnetic susceptibility and heat capacity[20,36] in the range of temperatures $10 \text{ K} \ll T_M \lesssim 100$ K. $Ce_3Bi_4Pd_3$, however, appears to have a significantly smaller Kondo gap, accompanied by an inverse residual resistivity ratio of only $R_{RRR}^{-1} \sim 5$ (see Fig. 2a) and maxima in the susceptibility (see Fig. 2b) and heat capacity (see Fig. 3) at $T_M \approx 5$ K. A possible explanation for the smaller Kondo gap in $Ce_3Bi_4Pd_3$ compared to its sister compound $Ce_3Bi_4Pt_3$ is the reduction in the strength of the hybridization identified in electronic structure calculations[37].

The small size of the Kondo gap in $Ce_3Bi_4Pd_3$ in Fig. 2a gives rise to a shallow slope of the resistivity Arrhenius plot in Fig. 2a (see Supplementary Fig. 2 for the magnetic field $B$-dependence), which therefore has a strong likelihood of being affected by the presence of in-gap states and changes in the transport scattering rate with temperature. To obtain an estimate of the Kondo gap that is less dependent on the scattering rate, we turn to the approximately activated behavior observed in Hall effect measurements[38] (see Fig. 4a), whereupon we obtain $\Delta = 1.8 \pm 0.5$ meV ($\approx 21$ K). This value is quantitatively consistent with estimates obtained from modeling the heat capacity and susceptibility (see below). An important factor in being able to extract the carrier density is that $\rho_{xy}$ (shown in Fig. 4a) is linear in $B$ at low $B$. We therefore find that $\Delta \approx k_B T_K$, which is quantitatively consistent with the hybridized many-body band

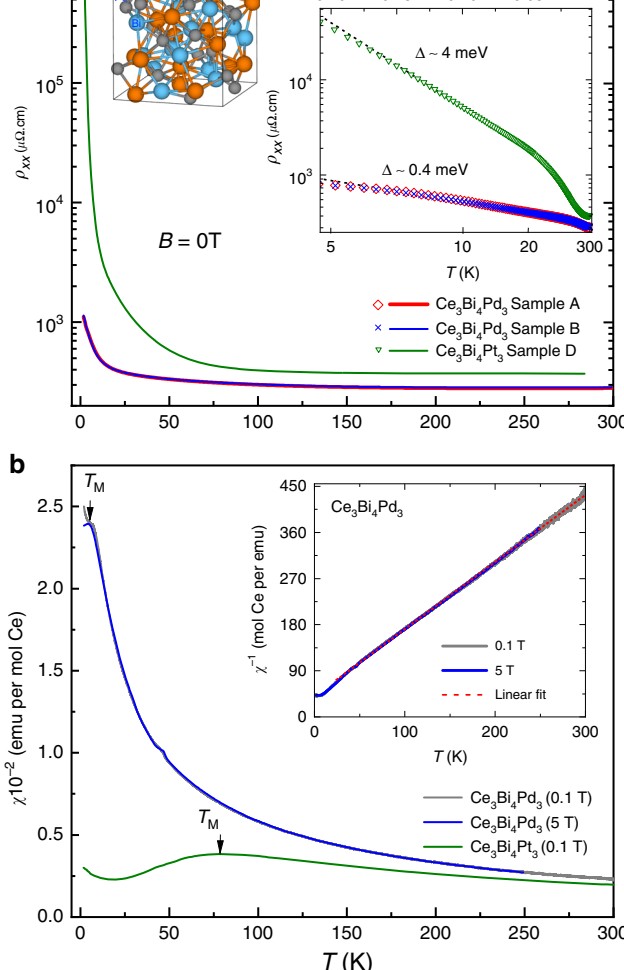

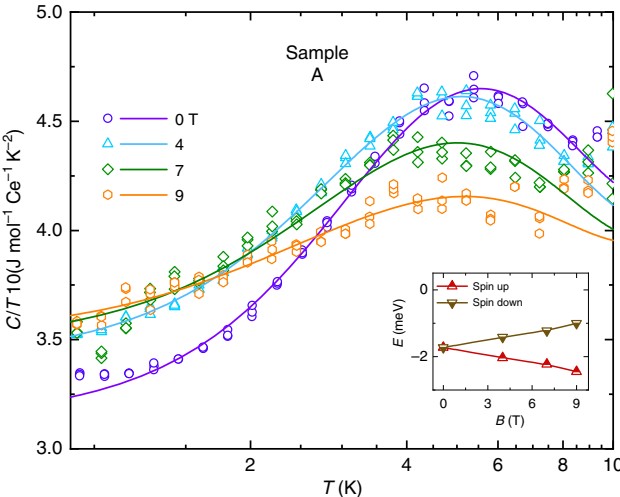

**Fig. 3** Specific heat data. Low-temperature heat capacity divided by temperature, $C/T$, plotted versus logarithmically scaled $T$ (symbols). Solid lines represent fits in which the Kondo gap is modeled using the Schotte–Schotte model (see Methods). A full fit is initially made only for $B = 0$ (purple line). On extending to fit to $C/T$ for $B = 4$, 7, and 9 T (light blue, green and yellow lines), only the Zeeman splitting $\pm\frac{1}{2}g_{eff}\mu_B B$ is allowed to vary (where $\mu_B$ is the Bohr magneton). The inset shows the locations of the Zeeman-split Lorentzians used in the Schotte–Schotte model (see Methods).

**Fig. 2** Sample characterization. **a** Comparison of $\rho_{xx}$ versus $T$ at $B = 0$ for two $Ce_3Bi_4Pd_3$ samples (labeled A and B) with $Ce_3Bi_4Pt_3$ (sample D). The inset graph shows Arrhenius plots. Whereas $Ce_3Bi_4Pt_3$ exhibits activated ($\rho_{xx} \propto e^{-\Delta/k_B T}$) behavior for $5 \lesssim T \lesssim 30$ K, $Ce_3Bi_4Pd_3$ appears to exhibit such behavior over a range $6 \lesssim T \lesssim 35$ K (with a smaller slope) that lies outside the fitting range, suggesting that $\Delta$ cannot be estimated reliably from $\rho_{xx}$ in the case of $Ce_3Bi_4Pd_3$. The shallow slope and departure from activated behavior at low $T$ is suggestive of a temperature-dependent scattering rate or residual in-gap states associated either with bulk defects, inclusions of other phases or the sample surface (see Methods). Also shown is a schematic of the crystal structure (see Methods). **b** Susceptibility $\chi$ versus $T$ for $Ce_3Bi_4Pd_3$ (at $B = 0.1$ and 5 T in gray and blue, respectively) compared with that for $Ce_3Bi_4Pt_3$ (at $B = 0.1$ T) from ref. [35], with $T_M$ indicating the maxima. The inset shows $1/\chi$ versus $T$ for $Ce_3Bi_4Pd_3$, evidencing Curie–Weiss behavior (indicated by red dashed fitted line) over a broad range of $T$ and a Curie constant $C \approx 0.8$ mol Ce per emu K.

picture[4]. The activated behavior can also be approximately rescaled against that measured in $Ce_3Bi_4Pt_3$ (see inset of Fig. 4a)[39] —the only significant difference being that $Ce_3Bi_4Pd_3$ has a several times smaller gap and a higher concentration of in-gap states, $n_0^*$ (corresponding to 0.2% holes per Ce), than $Ce_3Bi_4Pt_3$. Consistent with $Ce_3Bi_4Pd_3$ having a several times smaller gap, the insulating state is found to be suppressed by magnetic fields that are several times weaker in strength than those required in $Ce_3Bi_4Pt_3$ (see Fig. 4b).

The atypically small Kondo gap and finite concentration of in-gap states in $Ce_3Bi_4Pd_3$ imply that this system lies closer to the insulator-metal threshold[40] than archetypal Kondo insulators, leaving open the possibility that its Kondo insulating behavior is tuned by impurities and defects, as has been found to be the case in CeNiSn[41,42]. Although subsequent samples of CeNiSn prepared with fewer impurities were found to exhibit increased metallic behavior, the opposite has thus far been found in $Ce_3Bi_4Pd_3$[40,43]. Further, in nominally pure CeNiSn, a residual carrier density quantitatively similar to $n_0^*$ in the inset to Fig. 4a yields a coherent metallic state with an $a$-axis resistivity of between 20 and 50 $\mu\Omega$cm or less[42,44], whereas this same value of residual carrier density in $Ce_3Bi_4Pd_3$ yields an insulating behavior with an electrical resistivity that is 20 to 50 times higher. Indications are therefore that the residual carriers in $Ce_3Bi_4Pd_3$ are incoherent or extrinsic in contrast to CeNiSn (see Methods). Here, we consistently prepare samples of $Ce_3Bi_4Pd_3$ with insulating behavior at $B = 0$ (two examples of which are shown in Fig. 2) by polishing away surface oxidation (see Methods) and by avoiding superconducting binaries inherent to the flux technique used to grow these materials (see Supplementary Fig. 3). All samples prepared in such a manner have a thermally activated carrier density at low-magnetic fields (see Fig. 4a) of similar magnitude to that in $Ce_3Bi_4Pt_3$, and have an electrical resistivity that increases with decreasing $T$ for all $T \lesssim 250$ K (see Fig. 2a), which includes the absence of saturating behavior at low temperatures (see Methods and Supplementary Fig. 4)[45]. The above observations are consistent with the finding made by way of electronic structure calculations performed in the non-interacting limit (in which the $f$-electrons are regarded as itinerant) that $Ce_3Bi_4Pt_3$ and $Ce_3Bi_4Pd_3$ both similarly exhibit a charge gap[37,46], with the robustness of the Kondo gap being largely contingent upon the strength of $T_K$.

**Magnetic field-induced metallization.** Metallization of the Kondo insulating $Ce_3Bi_4Pd_3$ samples beyond $B_c \approx 11$ T (in the limit $T \to 0$) is evidenced in Figs. 1, 4b and 5 by the crossover

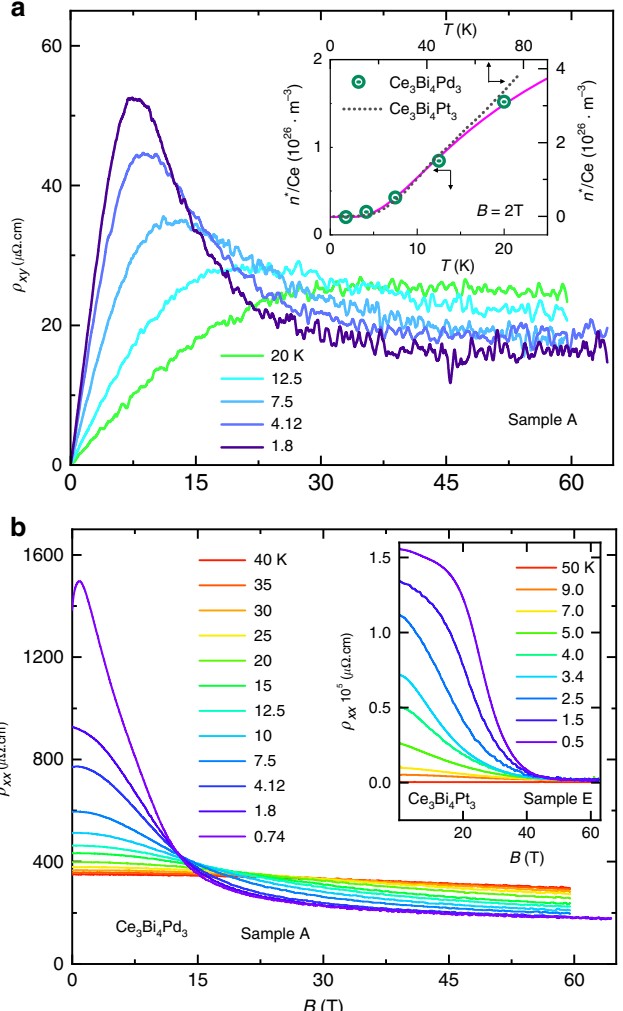

**Fig. 4** Magnetic field-dependent electrical transport measurements. **a** The Hall resistivity $\rho_{xy}$ of $Ce_3Bi_4Pd_3$ plotted versus $B$ measured at several temperatures. The inset shows the carrier density $n^*$ (green circles) plotted versus $T$ for $Ce_3Bi_4Pd_3$ at $B = 2$ T (left-hand and bottom axes), estimated using $n^* \approx B/e\rho_{xy}$ (not corrected for the anomalous Hall component), with the magenta line representing a fit to $n^* = n_0^* + n_1^* \exp(-\Delta/k_BT)$, where $n_0^* = 0.2 \times 10^{26}$ m$^{-3}$, and $n_1^* = 3.7 \times 10^{26}$ m$^{-3}$ and $\Delta = 1.8 \pm 0.5$ meV. The residual value of $n_0^*$ (equivalent to 2% of the Brillouin zone or ~0.2% per Ce) is smaller than the concentration of uncompensated Ce moments and magnetic impurities inferred from the susceptibility (see Methods), suggesting that the latter contribute only partly to the residual in-gap states. For comparison, the dotted line shows $n^*$ plotted versus $T$ for $Ce_3Bi_4Pt_3$ at $B = 5$ T (right-hand and top axes) from ref. [39]. **b** Measured magnetoresistivity $\rho_{xx}$ of $Ce_3Bi_4Pd_3$ sample A (for $B \parallel \langle 100\rangle$) at different temperatures (as indicated in different colors) plotted versus $B$. For comparison, the inset shows the magnetoresistance of $Ce_3Bi_4Pt_3$ at different temperatures.

from insulating behavior for $B < B_c$ to metallic behavior for $B > B_c$. We identify metallic behavior as an electrical resistivity having a value that decreases with decreasing temperature in the limit $T \to 0$ (i.e., $\partial\rho_{xx}/\partial T > 0$). Further evidence for the transition or crossover from insulating to metallic behavior with the closure of the Kondo insulating gap in $Ce_3Bi_4Pd_3$ is provided by Hall effect measurements at high-magnetic fields. The Hall resistivity $\rho_{xy}$ begins to drop on the approach to $B_c$, which indicates an increase in carrier density in advance of the closing of the Kondo gap, likely caused by thermal excitations of carriers across

the reduced gap. The nonlinearity of $\rho_{xy}$ above $B_c$ indicates the presence of multiple bands and a possible anomalous Hall contribution from skew scattering[47] (see Methods). In the absence of a dominant anomalous Hall component from skew scattering, the positive Hall coefficient implies either that the hole carriers have numerical supremacy over electron carriers, as would be expected on the basis of band structure calculations[37], or that the hole carriers are of higher mobility[38].

**Fermi liquid behavior**. On entering the high-magnetic field regime, we find the low-temperature electrical resistivity to display the Fermi liquid form $\rho_{xx} = \rho_0 + AT^2$, where $\rho_0$ is a constant and $A$ is a coefficient proportional to the square of the quasi-particle effective mass. The quadratic-in-temperature form is demonstrated by plotting the electrical resistivity versus $T^2$ in Fig. 5, yielding linear behavior at the lowest temperatures. The Fermi liquid temperature scale $T_{FL}$, which we define as the point beyond which the resistivity is observed to exhibit a downward curvature from $T^2$ behavior on increasing the temperature, is found to collapse on approaching $B_c$ in Fig. 1. At the same time, $A$ is observed to undergo a steep upturn on approaching $B_c$ from high-magnetic fields in Fig. 1. Both the upturn in $A$ and collapse of $T_{FL}$ are indicative of a collapsing Fermi liquid energy scale in the vicinity of $B_c$, which is one of the primary signatures of quantum criticality in Fermi liquid systems[48,49]. However, rather than peaking precisely at $B_c$, $A$ is observed to peak at $B \approx 15$ T. This is a likely consequence of the metallic contribution to the conductivity vanishing due to the collapse of the sizes of the Fermi surface pockets in the limit $B \to B_c$. Other possible factors include sample inhomogeneities causing $B_c$ to be non-uniform throughout the sample or an anisotropic gap causing the gap closure field to be momentum-space-dependent.

**Magnetic susceptibility measurements**. Thermodynamic evidence linking $B_c$ to the breakdown of the Kondo coupling accompanying the closing of the gap is provided by measurements of magnetization and magnetic susceptibility in Fig. 6. A strong argument for the reduction in Kondo screening is provided by the approach of the magnetization towards saturation in the inset to Fig. 6a, which is indicative of a state in which the $f$-electron moments are to a large extent polarized and subject to reduced fluctuations. More directly, we find the insulator-to-metal transition or crossover near $B_c$ to be accompanied by a low-temperature peak in the magnetic susceptibility $\chi$ in Fig. 6a, which we trace back to the peak in $\chi$ at $T_M$ along the temperature axis. On performing magnetic susceptibility measurements over a range of temperatures and magnetic fields, we construct a pseudo phase boundary from the locus of the peak that continuously connects $B_c$ with $T_M$ within the $B, T$ plane (see Fig. 1), revealing that $B_c = B(T_M \to 0)$. In the low-magnetic field limit, the broadness of the peak is indicative of a crossover, yet it becomes increasingly sharp on increasing the magnetic field and reducing the temperature.

**Specific heat measurements**. Further thermodynamic evidence for the magnetic field-induced closing of the Kondo gap is provided by the finding of a Schotte–Schotte anomaly in the specific heat in Fig. 3 (see also Supplementary Fig. 5), which undergoes a continuous change in shape in an applied magnetic field. A Schotte–Schotte[50,51] anomaly occurs under circumstances in which a peak in the electronic density-of-states in the shape of a Lorentzian is offset from the chemical potential by a gap (see Supplementary Fig. 6 and Methods). Fits (see Fig. 3a) to the heat capacity measured at $B = 0$ yield $\Delta = 1.7 \pm 0.2$ meV, which is in excellent agreement with $\Delta$ estimated from Hall effect

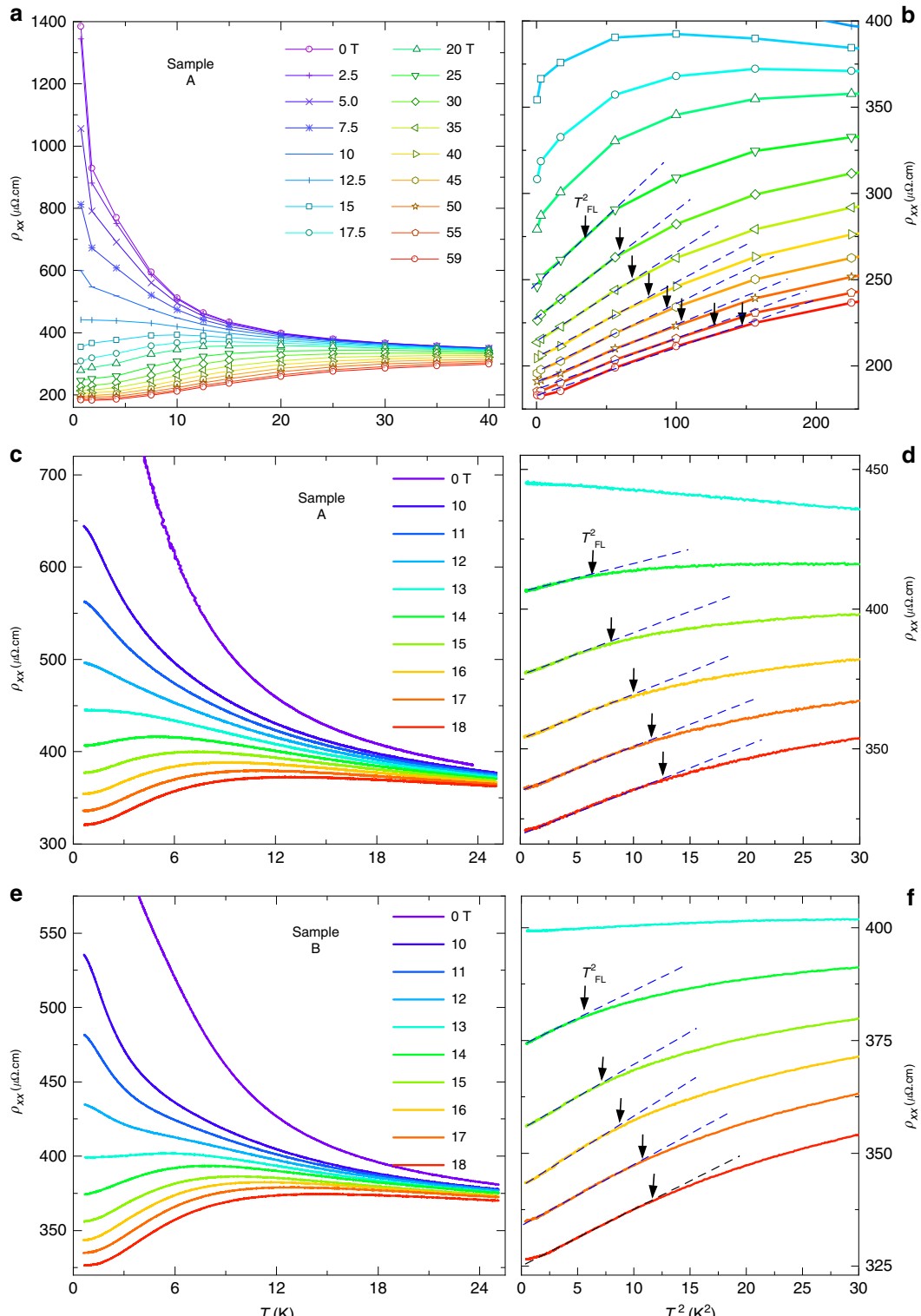

**Fig. 5** Temperature-dependent electrical resistivity measurements of Ce$_3$Bi$_4$Pd$_3$. **a** Electrical resistivity $\rho_{xx}$ versus $T$ at various values of the magnetic field (as indicated) extracted from Fig. 4a. Connecting lines are a guide to the eye. **b** $\rho_{xx}$ versus $T^2$ at various values of the magnetic field (as indicated) extracted from Fig. 4a. **c** $\rho_{xx}$ versus $T$ for the same sample measured at various steady static magnetic fields (as indicated). **d** The same $\rho_{xx}$ as in **c** versus $T^2$. **e** and **f** Same as for **c** and **d** but measured on a different sample (as indicated). Dashed lines indicate the linearity in $T^2$ while arrows indicate $T_{FL}$ (the temperature above which $\rho_{xx}$ appears to deviate from $T^2$ behavior). The slopes of the lines versus $T^2$ provide an estimate of $A$ (plotted in Fig. 1). A small departure from $T^2$ behavior in the limit $T^2 \rightarrow 0$ is sample-dependent, suggesting it to be of extrinsic origin.

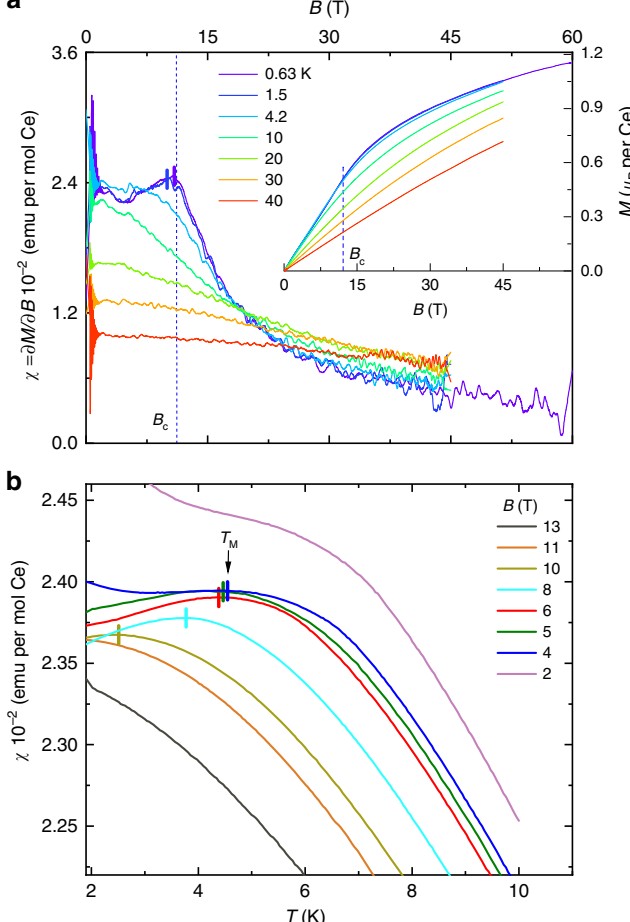

**Fig. 6** Magnetization and susceptibility measurements of $Ce_3Bi_4Pd_3$. **a** Magnetic susceptibility $\chi = \partial M/\partial B$ versus $B$ of $Ce_3Bi_4Pd_3$ (for $B \parallel \langle 100 \rangle$) at different temperatures (as indicated; see Methods). Vertical bars indicate the $T$-dependent maximum. The inset shows the magnetization $M$ versus $B$. **b** $\chi$ versus $T$ at different static magnetic fields (as indicated), with vertical bars indicating the maxima in the temperature dependence. Note that at $B = 2\,T$, a shoulder rather than a peak is observed due to a low $T$ contribution from magnetic impurities (or defects) at low $B$ and $T$ (see Methods). A similar low $T$ contribution is observed in $Ce_3Bi_4Pt_3$ in Fig. 2b, although, in that case, it is farther from $T_M$.

measurements (see Fig. 4a). On considering $B \neq 0$ and fixing all parameters except $g_{eff}$, we find that the evolution of the Schotte–Schotte anomaly under a magnetic field can be explained solely in terms of Zeeman splitting, where $g_{eff} \approx 2.9 \pm 0.3$ is an effective $g$-factor for pseudospins of $\pm\frac{1}{2}$. The critical magnetic field $B_c$ therefore roughly corresponds to $g_{eff}\mu_B B \approx \Delta \approx k_B T_K$. As a further consistency check, a similar value of the gap ($\Delta = 1.5 \pm 0.2$ meV) qualitatively accounts for the maximum in the magnetic susceptibility as a function of temperature (see Methods and Supplementary Fig. 7) and its shifting to lower temperatures under a magnetic field.

## Discussion

The signatures of strongly enhanced electronic correlations in the vicinity of the critical magnetic field $B_c \approx 11$ T in Figs. 1, 4, 5 and 6, and their association with an insulator-to-metal transition, are strikingly similar to those recently identified in $SmB_6$ as a function of pressure[28,31]. In $SmB_6$, the entry into the metallic phase with increasing pressure can be qualitatively understood in terms

of increased tendency for magnetism accompanying a continuous increase in the valence towards a trivalent value[27–31]. In $Ce_3Bi_4Pd_3$, the polarization of the $f$-electron moments by a magnetic field causes the magnetic field to behave like an effective negative pressure; the lattice volume invariably increases under a magnetic field in Ce compounds[52,53], causing a reduction in the valence towards a trivalent value. The relative ease by which $f$-electron moments are polarized by a magnetic field in $Ce_3Bi_4Pd_3$ (see inset to Fig. 6a) compared to $Ce_3Bi_4Pt_3$[54] suggests that the smaller Kondo temperature and gap in the former is a likely consequence of it lying closer to integer valence than the latter[37].

While a peak in the magnetic susceptibility at $B_c$ has been predicted to occur on closing the gap in a Kondo insulator[55], along with signatures of quantum criticality, other scenarios are possible at $B_c$, including the case found in itinerant electron metamagnets[56–58]. Here, quantum criticality may occur in the absence of broken symmetry due to the tuning of the highest temperature end point of a notional line of first order phase transitions to absolute zero[56]. Our finding of a peak in the susceptibility that becomes increasingly sharp on lowering the temperature suggests that $Ce_3Bi_4Pd_3$ remains in the crossover regime, however, with the end point remaining inaccessible in the limit $T \to 0$. It has yet to be determined whether this is an intrinsic property of the metal-insulator transition in $Ce_3Bi_4Pd_3$ or whether it is caused by the broadening of the quasiparticle bands by defects and impurities.

An alternative possibility, given the suppression of the Kondo coupling in strong magnetic fields, is that the Kondo coupling vanishes completely at $B_c$[59,60]—in effect causing the $f$-electrons to behave in a more localized fashion for $B > B_c$. Indeed, one of the predicted consequences of a localization of the $f$-electrons in $Ce_3Bi_4Pd_3$ (and $Ce_3Bi_4Pt_3$) is a metallic state[37]. The metallic state induced by the localization of the $f$-electrons has also recently been identified by way of dynamical mean field-theory as a strong candidate for a line-node semimetal[46], in which case $B_c$ would correspond to a insulator-to-line-node semimetal transition or, potentially, a topological quantum critical point[61]. The collapse of the Hall resistivity to a small value for $B \gtrsim B_c$ is indeed consistent with the Fermi liquid regime also being semimetallic.

The contiguity between a vanishing carrier density for $B < B_c$ and suppressed Kondo coupling for $B > B_c$ has the potential to make the nature of the magnetic field-tuned transition somewhat unique compared to that in other $d$- and $f$-electron systems. Whether the strong correlations are caused by proximity to a metamagnetic end point or a magnetic field-induced localization of the $f$-electrons, in both cases the reduction in electronic correlations beyond $B_c$ is generally expected to result in a gradual reduction in the effective mass $m^*$[57,62], or equivalently the $A$ coefficient[56,63], with increasing magnetic field. The reduction in $A$ with increasing field in $Ce_3Bi_4Pd_3$ is indeed qualitatively similar to that previously observed in $d$- and $f$-electron metamagnets (for example, $Sr_3Ru_2O_7$ and $CeRu_2Si_2$)[56,62,63].

Finally, there is the question of whether strong electronic correlations in proximity to a magnetic field-tuned quantum critical point contribute to the growth of the electronic contribution to the heat capacity in the vicinity of the field-induced transitions into the metallic states of the Kondo insulators $Ce_3Bi_4Pt_3$ and $YbB_{12}$[20,21], or to the temperature dependence of the quantum oscillation amplitudes within the insulating phases of $SmB_6$ and $YbB_{12}$[13,15]. Our findings suggest, at the very least, that a growth of electronic interactions at the expense of a declining hybridization needs to be taken into consideration in future more complete models of the insulating state in strong magnetic fields. A further possibility is that the Kondo insulator gradually transforms into an excitonic insulator[64] in advance of metallization.

## Methods

**Sample preparation.** $Ce_3Bi_4Pd_3$ crystallizes in a cubic structure of space group I43d with a noncentrosymmetric unit cell that contains four formula units and a lattice parameter of $a = 10.05$ Å[26]. The crystals are grown by the Bi-flux technique with starting composition Ce:Pd:Bi = 1:1:1.8. The reagents are put in an alumina crucible placed inside an evacuated quartz tube. The quartz tube is then heated to 1050 °C at 100 °C/h and is kept there for 8 h. The solution is then cooled down to 450 °C at 2 °C/h. The excess of Bi is removed by spinning the tube in a centrifuge. The crystallographic structure is verified by single-crystal diffraction at room temperature using Mo radiation in a Bruker D8 Venture diffractometer.

The upturn in the magnetic susceptibility at low temperature indicates the presence of uncompensated Ce moments and magnetic impurities (such as other rare earths)[1,3,32–34], which are frequently seen in Kondo insulators. However, inclusions of other crystalline phases such as $CePdBi_2$, which has a sharp antiferromagnetic transition at $T_N \approx 6$ K[65], and superconducting Bi-Pd binary compounds, which have a variety of different transition temperatures $T_c$, can also be present. The latter include tetragonal $PdBi_2$ ($T_c = 4.25$ K), BiPd ($T_c = 3.7$ K) and monoclinic $PdBi_2$ ($T_c = 1.73$ K). The absence of a sharp feature in the heat capacity at $\approx 6$ K rules out a significant presence of $CePdBi_2$. On fitting the magnetic susceptibility to a Kondo gap plus a low-temperature Curie–Weiss term to model uncompensated Ce moments and isolated magnetic impurities (see simulations of the magnetic susceptibility below), we estimate their concentration to be equivalent to ~2% per Ce. In the case of Tb impurities, which are determined from a chemical analysis of Ce to be the leading impurity, the larger moment would imply a significantly lower concentration of 0.1%. In samples that have been polished, no significant negative contribution to the susceptibility is found that would indicate a significant presence of superconducting impurity phases within the bulk, suggesting that superconductivity seen in electrical transport measurements occurs mostly on the surface (see electrical transport experimental details below).

At this early stage of sample growth, the electrical resistivity becomes higher with improved sample quality (see e.g., the difference between early work on these samples[40], this work and more recent work by another group[43]).

**Electrical transport experimental details.** Samples of $Ce_3Bi_4Pd_3$ are cut into rectangular shapes and polished for electrical transport measurements. All resistivity measurements, including those in static and pulsed magnetic fields are performed using the four wire technique. Whereas freshly polished samples (green line in Supplementary Fig. 4) exhibit insulating behavior down to $\approx 2$ K, thermally cycled samples exhibit additional superconductivity (red line in Supplementary Fig. 3). After re-polishing, most of the additional superconductivity is removed (blue line in Supplementary Fig. 3).

Despite the opening of a gap in the energy spectrum, Kondo insulators are prone to in-gap states that can reside either at the surface or the bulk and generally cause the electrical resistivity to acquire a finite value in the limit $T \rightarrow 0$. In a similar manner to $SmB_6$[45], $Ce_3Bi_4Pd_3$ is found to have an inverse resistivity $\rho_{xx}^{-1}$ that varies linearly in $T$ at low temperatures (see Supplementary Fig. 4), suggesting that scattering from defects may be responsible for the departure from activated behavior at the lowest temperatures (at least down to $\approx 2$ K)[45].

To delineate regions of the phase diagram where insulating $\partial \rho_{xx}/\partial T < 0$ and metallic $\partial \rho_{xx}/\partial T < 0$ are observed, the data in Fig. 5 are differentiated (see Supplementary Fig. 1) and the points where $\partial \rho_{xx}/\partial T = 0$ are plotted in Fig. 1.

**Magnetic susceptibility experimental details.** Magnetic susceptibility measurements in pulsed magnetic fields are performed using an extraction magnetometer[66]. Magnetic susceptibility measurements in static magnetic fields are performed using a Quantum Design vibrating sample magnetometer. Owing to isolated impurities giving rise to an upturn in the susceptibility at low temperatures[1,3,32–34], and the small energy scale associated with the Kondo gap in $Ce_3Bi_4Pd_3$, a steady magnetic field of $B \gtrsim 4$ T is required to uncover the maximum in the susceptibility. Only a shoulder is observed at lower magnetic fields (see simulations of the magnetic susceptibility below). In a similar manner to $Ce_3Bi_4Pt_3$[54], the magnetic susceptibility of $Ce_3Bi_4Pd_3$ continues to remain large as the magnetic field is increased, which is suggestive of a band Van Vleck paramagnetism effect[23].

**Heat capacity experimental and modeling details.** Heat capacity measurements on $Ce_3Bi_4Pd_3$, made using a Quantum Design Physical Properties Measuring System, reveal a peak in the overall heat capacity centered at $\approx 5$ K (see Supplementary Fig. 5). In $Ce_3Bi_4Pt_3$, by contrast, a peak is observed in the heat capacity centered at $\approx 50$ K[20,23]. The heat capacity is found to be reproducible in different samples for $T \gtrsim 1.5$ K. Features that appear below $\approx 1.5$ K are sample-dependent, suggesting them to be of extrinsic origin.

A Schotte–Schotte anomaly occurs when a peak in the electronic density-of-states in the shape of a Lorentzian is offset by an energy gap $\Delta$ relative to the chemical potential (see Supplementary Fig. 6a). In the single impurity limit, the width $2W$ of the Lorentzian is compared to the Kondo temperature. However, it will also include the effect of quasiparticle scattering from nonmagnetic defects and impurities. While earlier it had been assumed that the Kondo gap is symmetric

with respect to the chemical potential (at $\varepsilon = 0$)[20], band structure calculations of $Ce_3Bi_4Pt_3$ and $Ce_3Bi_4Pd_3$ have indicated an asymmetric gap with flat bands situated just below the chemical potential[37]. Flat bands are also situated above the chemical potential, but are considerably further away in energy. Given what is known about the electronic structure, we model the Schotte–Schotte anomaly using an electronic density of states of the form

$$D(\varepsilon) = D_0 \sum_{\sigma = \pm \frac{1}{2}} \frac{W}{\pi(W^2 + (\varepsilon + \Delta + \sigma g_{\text{eff}} \mu_B B)^2)}, \tag{1}$$

where $\sigma g_{\text{eff}} \mu_B B$ represents Zeeman splitting. Here, $\sigma$ represents pseudospins of $\pm \frac{1}{2}$ while $g_{\text{eff}}$ is an effective $g$-factor. The free energy given by $F = -k_B T \int_{-\infty}^{\infty} D(\varepsilon) \ln[1 + \exp(-\varepsilon/k_B T)] d\varepsilon$ while the heat capacity given by $C_p \approx C_v = T \partial^2 F/\partial T^2|_v$, which we calculate numerically[67] and fit to the experimental data over the temperature range $1.5 \lesssim T \lesssim 10$ K. During fitting, we assume a phonon contribution $C_{ph}/T = \beta T^2$ where $\beta = 1$ mJ mol$^{-1}$ Ce$^{-1}$ K$^{-4}$. It is important to note, that on fitting to the Schotte–Schotte model, one cannot distinguish whether $\Delta$ is positive (corresponding to a peak below the chemical potential) or whether it is negative (corresponding to a peak above the chemical potential), or whether there are peaks located at both $\pm\Delta$ as has sometimes been assumed[20].

We initially fit the data only at $B = 0$, from which we obtain $\Delta = 1.7 \pm 0.2$ meV, which is comparable to the value estimated from Hall effect measurements (see Fig. 4a), and $W = 1.1 \pm 0.2$ meV. For $B \neq 0$, we hold $D_0$, $W$, and $\Delta$ constant and allow only $g_{\text{eff}}$ to vary. Zeeman splitting alone, with $g_{\text{eff}} = 2.9 \pm 0.3$, is able to explain the evolution of the Schotte–Schotte peak under a magnetic field. $\Delta - W \approx 0.6$ meV ($\approx 7$ K) is of comparable order to the reduced gap estimated from the slope of the Arrhenius plot in the inset to Fig. 2a.

A distinguishing feature of the Schotte–Schotte anomaly that appears to be present in the experimental data (see Fig. 3 and ref. [40]) is an electronic contribution to the heat capacity of the form

$$C_{el}/T \approx \gamma_0 + (\beta' - \beta)T^2 \tag{2}$$

(see Supplementary Fig. 5b) in the limit $T \rightarrow 0$, where $(\beta' - \beta)T^2$ exceeds the contribution $\beta T^2$ from phonons and where $\gamma_0$ is the Sommerfeld coefficient characterizing in-gap states (that also depends on the strength of the magnetic field)[67]. Supplementary Fig. 5b shows that $Ce_3Bi_4Pd_3$ exhibits the form given by Eq. (2) in the limit $T \rightarrow 0$, with $\gamma_0$ increasing by $\approx 30$ mJmol$^{-1}$K$^{-2}$ on increasing the magnetic field from 0 to 4 T. A comparable increase was reported in $Ce_3Bi_4Pt_3$ on increasing the magnetic field from 0 to 40 T[20].

The observed consistency with the Schotte–Schotte model therefore includes (i) a peak at $\approx 5$ T, (ii) a residual electronic contribution $\gamma_0$ and (iii) a magnetic field-dependent low- temperature behavior of the form $C_{el}/T \approx \gamma_0 + (\beta' - \beta)T^2$ (see Supplementary Fig. 5 and Methods). It remains to be determined to what extent the Lorentzian line-shape (of halfwidth $W = 1.1 \pm 0.2$ meV) is intrinsic (e.g., related to the Kondo coupling and electronic bandwidths) or affected by quasiparticles scattering from defects and impurities.

**Simulations of the magnetic susceptibility.** As a further consistency check, we show that similar parameters as used in the modeling of the heat capacity also provide a qualitative explanation for the approximate location of the peak in the susceptibility as a function of temperature and its suppression to lower temperatures as the magnetic field is increased. We calculate the susceptibility using

$$\chi = C \int_{-\infty}^{\infty} D(\varepsilon) f'_{\text{FD}}(\varepsilon) d\varepsilon, \tag{3}$$

(valid for low-magnetic fields) where $f'_{\text{FD}}(\varepsilon)$ is the derivative of the Fermi-Dirac distribution function and $C$ is a constant renormalized to the experimental susceptibility at 300 K, the results of which are plotted in Supplementary Fig. 7. Equation (3) does not include the low-temperature Curie–Weiss-like contribution from impurities and defects. Partly for this reason, it overestimates the the initial growth in the susceptibility with increasing field. The model performs poorly once $B \sim B_c$, however, which is likely a consequence of electronic correlations not being included. Electrical transport measurements indicate electronic correlations to be important in the vicinity of $B_c$. On comparing the calculated peaks in the susceptibility ($B < B_c$) with those obtained in Fig. 6b, we estimate $\Delta = 1.5 \pm 0.2$ meV and $g_{\text{eff}} = 2.7 \pm 0.2$ from the susceptibility.

To obtain a rough estimate of the concentration of uncompensated Ce moments and magnetic impurities, we fit the low-temperature portion of the magnetic susceptibility at 0.1 T (inset to Supplementary Fig. 7) to $\chi_{\text{fit}} = (1 - f)\chi + f\chi_{\text{imp}}$, where $f$ is the volume fraction of impurities, $\chi$ is the susceptibility given by Eq. (3) and $\chi_{\text{imp}}$ corresponds to the impurity concentration. We model this by assuming a modified version of $\chi$ in which $D(\varepsilon)$ is replaced by a delta function (for simplicity, we assume the total number of states $\int_{-\infty}^{\infty} D(\varepsilon) d\varepsilon$ and effective $g$-factor are the same as used in modeling the Kondo gap). We estimate $f \sim 2\%$.

**Anomalous Hall effect estimate**. We estimate the anomalous Hall effect contribution from skew scattering within the high-magnetic field metallic phase using $\rho_{xy}^A = \gamma \tilde{\chi} \rho_{xx}$[47], where $\gamma \sim 0.08\,KT^{-1}$ is a typical value of the coefficient for Ce compounds[47]. At $B = 0$, the reduced susceptibility is $\tilde{\chi} \approx \chi_{5T}^{peak}/C \approx 0.03\,K^{-1}$, but this falls to $\approx 0.014\,K^{-1}$ at 60 T. Using $\rho_{xx} \approx 160\,\mu\Omega$cm at 60 T, we obtain $\rho_{xy}^A \sim 1.5 \times 10^{-8}\,\Omega$m at 60 T, which is $\sim 10\%$ of $\rho_{xy}$.

## Data availability

The data presented in this manuscript is available from the corresponding author on reasonable request.

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

## Acknowledgements

This high-magnetic field work was supported by the US Department of Energy "Science of 100 tesla" BES program. S.K.K. acknowledges support of the LANL Directors Post-doctoral Funding LDRD program. Sample characterization at Los Alamos National Laboratory was performed under the auspices of the U.S. Department of Energy, Office of Basic Energy Sciences, Division of Materials Sciences and Engineering, "Quantum Fluctuations in Narrow Band Systems" project. The high-magnetic field facilities of the National High-Magnetic Field Laboratory—PFF facility and related technical support are funded by the National Science Foundation Cooperative Agreement Number DMR-1157490 and DMR-1644779, the State of Florida and the U.S. Department of Energy. We acknowledge helpful discussions with Mucio Continentino, Peter Riseborough, Chao Cao, Jianxin Zhu, Qimiao Si. and Peter Wölfle. M.K.C. acknowledges support by the Laboratory Directed Research and Development program of Los Alamos National Laboratory under project number 20180137ER in conducting experiments on Kondo insulators. N.H. acknowledges support from the Los Alamos National Laboratory LDRD program: project number 20180025DR.

## Author contributions

S.K.K., M.K.C., J.P., S.M.T., and N.H. performed the measurements. P.F.S.R. and E.D.B. grew and characterized the samples. S.K.K., N.H., and P.F.S.R. wrote the manuscript. J.D.T. and F.R. assisted with the interpretation of the experimental data.

## Competing interests

There are no competing interests.
