## [Peer Review File · Nature Communications]

Reviewers' comments:

Reviewer #1 (Remarks to the Author):

This is a very interesting paper studying the closing of the gap in the small gap Kondo insulator $\text{Ce}_3\text{Bi}_4\text{Pd}_3$. I find the paper particularly intriguing, as the results suggest that there is an order parameter involved in the insulator to metal transition in this compound. I strongly suggest the publication of the manuscript, after the authors address the following issues:

- 1) They establish a phase diagram which looks very much like the canonical phase diagram for a QCP due to the suppression of some order parameter. I think the authors should stress this fact and discuss in more detail what such an order parameter could be. There are some pointers given to the literature, but it is dispersed throughout the manuscript.
- 2) In Fig. 2 I have a hard time seeing the data shown in red. Also, the data shown in green is the Pt version, whereas the other samples are indicated as A and B of actually the Pd version of the compound. This is confusing. The caption is not great either.
- 3) Fig. 9 b: The solid lines should be explained in the caption.
- 3) For FeSi the activation gap can be extracted from the mag. susceptibility. Since they have the data, why not try this fit?

Reviewer #2 (Remarks to the Author):

Referee Report Kushwaha et al.

The paper by Kushwaha et al reports measurements of resistivity, Hall effect, magnetization and specific heat of the Kondo insulator $\text{Ce}_3\text{Bi}_4\text{Pd}_3$, showing that the system is entering a field-induced metallic state. The experimental data are of high quality and analysis and discussion at a high level.

Overall, I think the results are definitely worth publishing in Nature Communications since the system indeed presents a very good opportunity to study the insulator-metal transition in Kondo insulators at accessible magnetic fields. Therefore, the paper is of general interest to the community and also to the wider field of physics. However, in my opinion, the authors overdo the interpretation of their data by claiming that the insulator-metal transition is happening through a quantum critical point. It is an attractive possibility, but the data are not clear enough to support this point unequivocally. Even without this claim the manuscript is worth publishing. Since it is one of the main claims of the paper, some revisions and more careful wording will be needed to reformulate the main findings.

Please find below my concerns and minor comments explained in detail.

1. Title: Please remove/reformulate the second part.
2. Information on crystal structure is completely missing. How do the Bi-Pd impurity phases influence the results? What percentage of the crystal do they present? Please extend the methods section to include what is known about crystallographics and impurities in this and similar systems.
3. Crystal quality: The resistance increase with lowering temperature of $\text{Ce}_3\text{Bi}_4\text{Pd}_3$ is rather small compared to other Kondo insulators. Since the gap is small, the system might be quite sensitive to impurities, as also mentioned in the manuscript and indicated by the larger density of in-gap states found in the Hall data. As well, the field-induced decrease of the resistivity is again small compared to the pressure induced one in SmB_6 . The authors should comment more explicitly on sample quality and how this could affect their results such as the broadness of the transitions. The

absence of quantum oscillations at high field might also be an indication of rather bad sample quality. Refer to methods section.

4. Figure 1 and Fig. 5: The points at B_c / T_M suggest sharp transitions or even a phase transition, since the authors even talk about a quantum critical point, where a phase transition is usually a prerequisite. However, when I look at the data, there are no sharp transitions. The sharpest features are given in the Hall resistivity versus field and the susceptibility versus field at the two lowest temperature. But even those are not sharp enough for a phase transition in any way. The resistivity versus field shows no sharp feature. Also, the field scales from the different probes are slightly different. The broadness of the transition should be made visual in Figure 1. The theory of reference 10 might apply here, but I don't see any indication of a first order transition in the data either. Why should this model apply here? This seems a possibility but given the quality of the sample/data, it is a bit too far-fetched. Do the authors believe that crystal impurities hide sharp phase transitions here?

The authors should formulate their claims with respect to the possibility of a quantum critical point more carefully in the title, abstract and the main manuscript.

5. Fig. 5a: In the vertical bars in (a) indicate a B-dependent maximum. The legend says T-dependent. The bar of the 4.2 K curve should be removed since the data is saturating rather than going through a maximum. Please refer to the methods section for the low-T contribution.

6. Fig. 1 and Fig. 4: The definition of TFL is rather unclear to me. I would like to see curves of data – fit versus T^2 for the different curves in Fig 4 d and f to be convinced that there is a well-defined Fermi liquid behavior anywhere and a departure from it. If found appropriately, the error bars on TFL should be adapted.

7. I would like to see a plot of gap versus field from the resistivity data as was done in SmB6 as a function of pressure. This would be more convincing evidence for a sharp transition. The fact that the gap temperature lies just outside the fitting range where ρ shows activated behavior is an argument against this fit. How does this behave with field?

8. Fig. 6: remove the "at $B = 0$ " in the legend.

9. Labels in the figures are too small. Fit lines are not visible and not possible to follow / compare with the data.

10. The numbering of the references is not in order in the abstract. 11 appears before 9 and 10.

11. Fig. 4 c and e: The symbols of the data could be removed since they are too small to be visible. Colored lines are just fine.

Reviewer #3 (Remarks to the Author):

This paper entitled by "Metallization of a Kondo insulator at a quantum critical point" reports the measurements of magnetic susceptibility, electrical resistance, Hall resistance, and specific heat of $\text{Ce}_3\text{Bi}_4\text{Pd}_3$ in magnetic fields. The authors interpret the change in the resistivity vs T^2 curve as the evidence for the transition from a Kondo insulating state to a heavy Fermi liquid state.

Furthermore, the authors argue that the plausible coincidence of the suppression of T_M and the development of T_{FL} indicates a quantum critical point. Although the experimental results seem to be reliable, the analysis and presentation of the data are not sufficiently careful to obtain scientific conclusions. Therefore, I do not recommend this paper to be published in Nature Communications. I mention the following problems. Because some of them are primitive, I suspect that not all the authors read the manuscript carefully.

1. The authors classify $\text{Ce}_3\text{Bi}_4\text{Pd}_3$ as a Kondo insulator without providing experimental evidence expect for the 2-3 times increase in the electrical resistivity on cooling from 24 K to 2 K. The large values of the residual carrier density of $0.2 \times 10^{26} \text{ m}^{-3}$ and C/T of $0.32 \text{ Jmol}^{-1}\text{Ce}^{-1}\text{K}^{-2}$ (FIG.9) are at variance with an insulating state. Instead, such large values of carrier density and C/T are characteristic of Kondo semimetals. For a Kondo semimetal CeNiSn , application of magnetic field closes the pseudogap in the density of states, giving rise to a negative magneto resistance. See

e.g., K. Sugiyama et al., J. Phys. Soc. Jpn. 67, 2455 (1998).

2. In Fig. 1, it is noted that the data of TM are obtained from the maximum in $\chi_{\text{ai}}(T)$ (triangles) and B (stars) from Fig. 5. There is a maximum in χ_{ai} (B, T= 0.63 K) at around 11 T while there is no maximum in χ_{ai} (B, T=1.5 K). Therefore, I do not trust the star plotted in Fig. 1.

3. In the inset of Fig. 2a, the unit in the right-hand-axis is lacking. "Ce₃Bi₄Pd₃" on the top should be deleted because the data for Ce₃Bi₄Pt₃ are also plotted in this figure. In Fig. 2-6, the description of the scale such as 10^{-2} (emu/mol.Ce) does not follow the standard style.

4. The slope of the χ_{ai}^{-1} vs T data in the inset of Fig. 2b changes abruptly at around 250 K. The authors are requested to explain the reason.

5. In Fig. 3a, the thermal activation energy was estimated from a fit to the three data points of $n^*(T)$ at T = 7.5, 12.5, and 20 K. To obtain the activation energy in the Hall coefficient, I suggest the authors to present the Arrhenius plot of $\log R_{\text{H}}(T)$ vs $1/T$.

6. In Fig. 4, the $\rho(T)$ data in the fields of 12.5 -13 T become constant for T < 6 K. However, this field range is included in the FL state in Fig. 1.

7. The authors used the Schotte-Schotte formula to analyze the data of C/T under constant magnetic fields up to 9 T. It is necessary to show and analyze the data for B > B_c = 11 T to confirm the change in the density of states associated with the gap closure. For a Kondo semimetal, such analysis was reported by K. Izawa et al., PRB. 59, 2599 (1999).

Our response to the Reviewer comments are indicated below in **bold red font**. Changes to the main text (other than trivial typos) are highlighted in **violet**.

Reviewer #1 (Remarks to the Author):

This is a very interesting paper studying the closing of the gap in the small gap Kondo insulator $\text{Ce}_3\text{Bi}_4\text{Pd}_3$. I find the paper particularly intriguing, as the results suggest that there is an order parameter involved in the insulator to metal transition in this compound. I strongly suggest the publication of the manuscript, after the authors address the following issues:

1) They establish a phase diagram which looks very much like the canonical phase diagram for a QCP due to the suppression of some order parameter. I think the authors should stress this fact and discuss in more detail what such an order parameter could be. There are some pointers given to the literature, but it is dispersed throughout the manuscript.

Response to (1): We agree with the referee that our phase diagram resembles the canonical phase diagram of a QCP generated by the suppression of some order parameter. Nevertheless, we remind the referee that the presence of an order parameter and symmetry breaking are not necessary conditions for the emergence of quantum critical phenomena. Topological quantum phase transitions and Kondo breakdown QCPs associated with a vanishing Kondo scale are well-known examples [References 60 & 61 of the manuscript]. We therefore refer to the peaks/maxima in the susceptibility of $\text{Ce}_3\text{Bi}_4\text{Pd}_3$ as a “pseudo phase boundary” because neither a phase transition nor a symmetry breaking is identified. The observed features are (i) the peak in the susceptibility, (ii) the onset of Fermi liquid behavior and (iii) the crossover from insulator to metal behavior (this has been added; see below). The relation between the experimental data and several different QCP scenarios is examined in the Discussion section.

2) In Fig. 2 I have a hard time seeing the data shown in red. Also, the data shown in green is the Pt version, whereas the other samples are indicated as A and B of actually the Pd version of the compound. This is confusing. The caption is not great either.

Response to (2): The red line is now thicker so that it is clearly visible behind the blue line, and the red and blue symbols have been changed so that both are identifiable. The labeling of the various lines with regards to $\text{Ce}_3\text{Bi}_4\text{Pd}_3$ and $\text{Ce}_3\text{Bi}_4\text{Pt}_3$ has also been clarified.

3) Fig. 9 b: The solid lines should be explained in the caption.

Response to (3a): The lines and symbols are now labeled. The figure is now Supplementary Figure 5b.

3) For FeSi the activation gap can be extracted from the mag. susceptibility. Since they have the data, why not try this fit?

Response to (3b): Owing to the additional Curie Weiss contribution from impurities at low B and T, we have calculated the susceptibility using parameters similar to those used for the heat capacity fits. The results are shown in Supplementary Figure 7. At B=0.1 T we have performed a low temperature fit to the susceptibility that includes a Curie Weiss contribution from impurities. The purpose of the latter is solely to estimate the concentration of magnetic impurities. The details are discussed in the Methods.

Reviewer #2 (Remarks to the Author):

The paper by Kushwaha et al reports measurements of resistivity, Hall effect, magnetization and specific heat of the Kondo insulator Ce₃Bi₄Pd₃, showing that the system is entering a field-induced metallic state. The experimental data are of high quality and analysis and discussion at a high level.

Overall, I think the results are definitely worth publishing in Nature Communications since the system indeed presents a very good opportunity to study the insulator-metal transition in Kondo insulators at accessible magnetic fields. Therefore, the paper is of general interest to the community and also to the wider field of physics. However, in my opinion, the authors overdo the interpretation of their data by claiming that the insulator-metal transition is happening through a quantum critical point. It is an attractive possibility, but the data are not clear enough to support this point unequivocally. Even without this claim the manuscript is worth publishing. Since it is one of the main claims of the paper, some revisions and more careful wording will be needed to reformulate the main findings.

Please find below my concerns and minor comments explained in detail.

1. Title: Please remove/reformulate the second part.

Response to (1): The title has been changed so that it emphasizes the emergence of Fermi liquid behavior at high fields.

2. Information on crystal structure is completely missing. How do the Bi-Pd impurity phases influence the results? What percentage of the crystal do they present? Please extend the methods section to include what is known about crystallographics and impurities in this and similar systems.

Response to (2): More information regarding the crystal structure and impurity phases has been added to the Methods section.

3. Crystal quality: The resistance increase with lowering temperature of Ce₃Bi₄Pd₃ is rather small compared to other Kondo insulators. Since the gap is small, the system might be quite sensitive to impurities, as also mentioned in the manuscript and indicated by the larger density of in-gap states found in the Hall data. As well, the field-induced decrease of the resistivity is again small compared to the pressure induced one in SmB₆. The authors should comment more

explicitly on sample quality and how this could affect their results such as the broadness of the transitions. The absence of quantum oscillations at high field might also be an indication of rather bad sample quality. Refer to methods section.

Response to (3): We have added more details pertaining to sample quality and the broadness of the transition. This includes (i) a discussion in the Metallization subsection about the width of the transition and the 1.5T difference between the peak in the susceptibility and the field at which $dp_{xx}/dT = 0$ (i.e. the metal-insulator transition seen in electrical transport), (ii) an estimate of the concentration of magnetic impurities (e.g. Tb, or uncompensated Ce sites) in the Methods section, (iii) an estimate of the residual carrier density (of about 0.2 holes per Ce) in the first subsection of the Results, and (iv) more discussion relating the QCP to broad transitions in the Discussion section.

4. Figure 1 and Fig. 5: The points at B_c / T_M suggest sharp transitions or even a phase transition, since the authors even talk about a quantum critical point, where a phase transition is usually a prerequisite. However, when I look at the data, there are no sharp transitions. The sharpest features are given in the Hall resistivity versus field and the susceptibility versus field at the two lowest temperature. But even those are not sharp enough for a phase transition in any way. The resistivity versus field shows no sharp feature. Also, the field scales from the different probes are slightly different. The broadness of the transition should be made visual in Figure 1. The theory of reference 10 might apply here, but I don't see any indication of a first order transition in the data either. Why should this model apply here? This seems a possibility but given the quality of the sample/data, it is a bit too far-fetched. Do the authors believe that crystal impurities hide sharp phase transitions here?

The authors should formulate their claims with respect to the possibility of a quantum critical point more carefully in the title, abstract and the main manuscript.

Response to (4): We agree with the referee that our phase diagram resembles the canonical phase diagram of a QCP generated by the suppression of some order parameter. Nevertheless, we remind the referee that the presence of an order parameter and symmetry breaking are not necessary conditions for the emergence of quantum critical phenomena. Topological quantum phase transitions and Kondo breakdown QCPs associated with a vanishing Kondo scale are well-known examples [References 60 & 61 of the manuscript]. We therefore refer to the peak in the susceptibility of $Ce_3Bi_4Pd_3$ as a "pseudo phase boundary" because neither a phase transition nor a symmetry breaking is identified.

We have added a line (and datapoints) at which $dp_{xx}/dT = 0$ so as to indicate the crossover between insulating and metallic behavior. This helps to clarify its relation to the peak in the susceptibility.

Indeed, we find no evidence for a first order phase transition, but we remind the referee that, in the weak coupling regime, Reference [10] (now Ref [55]) does predict a *continuous* metal-

insulator transition. To be as comprehensive as possible, we do consider other possibilities in the Discussion section, such as the proximity to an endpoint or breakdown of the Kondo coupling. Broadening due to scattering or sample inhomogeneity could certainly be a factor; both possibilities of which we now explicitly refer to in the text. We emphasize, however, that as we approach zero temperature, the observed features do become much sharper. Nevertheless, the claim of a QCP is made somewhat more tentatively throughout the manuscript to comply with the referee's request. The experimental data appear to mostly support a scenario in which a breakdown in the Kondo coupling occurs in the vicinity of B_c [Reference 60 in the revised manuscript].

5. Fig. 5a: In the vertical bars in (a) indicate a B-dependent maximum. The legend says T-dependent. The bar of the 4.2 K curve should be removed since the data is saturating rather than going through a maximum. Please refer to the methods section for the low-T contribution.

Response to (5): The 4.2 K bar in Fig. 5a has now been removed as well as the corresponding datapoint in Fig. 1. The Methods section is now referred to in the caption.

6. Fig. 1 and Fig. 4: The definition of TFL is rather unclear to me. I would like to see curves of data – fit versus T^2 for the different curves in Fig 4 d and f to be convinced that there is a well-defined Fermi liquid behavior anywhere and a departure from it. If found appropriately, the error bars on TFL should be adapted.

Response to (6): The T^2 fit lines were previously obscured from view by large data points. This issue has been fixed in the updated version of the manuscript.

7. I would like to see a plot of gap versus field from the resistivity data as was done in SmB6 as a function of pressure. This would be more convincing evidence for a sharp transition. The fact that the gap temperature lies just outside the fitting range where ρ shows activated behavior is an argument against this fit. How does this behave with field?

Response to (7): We have added Supplementary Fig. 2 with such a plot, which is also referred to in the Results section.

8. Fig. 6: remove the "at B = 0" in the legend.

Response to (8): The caption has been revised so that it more accurately describes the fitting procedure.

9. Labels in the figures are too small. Fit lines are not visible and not possible to follow / compare with the data.

Response to (9): We have improved the clarity of fit lines and labels.

10. The numbering of the references is not in order in the abstract. 11 appears before 9 and 10.

Response to (10): The ordering of the references has been revised.

11. Fig. 4 c and e: The symbols of the data could be removed since they are too small to be visible. Colored lines are just fine.

Response to (11): The symbols have been changed to lines as requested.

Reviewer #3 (Remarks to the Author):

This paper entitled by “Metallization of a Kondo insulator at a quantum critical point” reports the measurements of magnetic susceptibility, electrical resistance, Hall resistance, and specific heat of Ce₃Bi₄Pd₃ in magnetic fields. The authors interpret the change in the resistivity vs T^2 curve as the evidence for the transition from a Kondo insulating state to a heavy Fermi liquid state. Furthermore, the authors argue that the plausible coincidence of the suppression of T_M and the development of T_{FL} indicates a quantum critical point. Although the experimental results seem to be reliable, the analysis and presentation of the data are not sufficiently careful to obtain scientific conclusions. Therefore, I do not recommend this paper to be published in Nature Communications. I mention the following problems. Because some of them are primitive, I suspect that not all the authors read the manuscript carefully.

1. The authors classify Ce₃Bi₄Pd₃ as a Kondo insulator without providing experimental evidence expect for the 2-3 times increase in the electrical resistivity on cooling from 24 K to 2 K. The large values of the residual carrier density of $0.2 \times 10^{26} \text{ m}^{-3}$ and C/T of $0.32 \text{ J mol}^{-1} \text{ K}^{-2}$ (FIG.9) are at variance with an insulating state. Instead, such large values of carrier density and C/T are characteristic of Kondo semimetals. For a Kondo semimetal CeNiSn, application of magnetic field closes the pseudogap in the density of states, giving rise to a negative magnetoresistance. See e.g., K. Sugiyama et al., J. Phys. Soc. Jpn. 67, 2455 (1998).

Response to (1): This is a valid concern, and we have therefore revised the 3rd paragraph of the results section to address this issue (we also cite some CeNiSn references). While there is a finite possibility that Ce₃Bi₄Pd₃ could turn out to be semimetal, there are arguments based on our data for why this would not be the case:

- (i) Though we are still at early stages of sample growth, there have been already improvements in the sample quality, which yield more insulating behavior in Ce₃Bi₄Pd₃. The electrical resistivity becomes higher with improved sample quality. See, for example, the difference between early work on these samples [Reference 40 in the revised manuscript] and more recent work by the same group [Reference 43]. Our resistivity data are similar to those in Ref. 43. This is in contrast to CeNiSn, for which improved sample quality yielded more metallic behavior [see References 42 and 44 of the revised manuscript].**

- (ii) **The residual carrier density we find for Ce₃Bi₄Pd₃ is the same as that reported in CeNiSn [Reference 44 of the revised manuscript]. However, CeNiSn becomes metallic at this concentration (with a mobility >4000 cm²/Vs), whereas Ce₃Bi₄Pd₃ continues to exhibit insulating behavior (with a mobility <200 cm²/Vs), suggesting that a coherent Fermi liquid state does not form at B=0 in Ce₃Bi₄Pd₃. We cannot exclude such a possibility occurring at much smaller residual carrier densities in Ce₃Bi₄Pd₃, although such samples – if attainable - have not yet been synthesized.**
- (iii) **The magnetoresistance of CeNiSn in the vicinity of B_c according to the above reference is an order of magnitude smaller than that of Ce₃Bi₄Pd₃. At B=0, the a-axis resistivity is ~10² times smaller than that of Ce₃Bi₄Pd₃. For this reason, the CeNiSn measurement of Sugiyama et al does not appear to be similar to what we find in Ce₃Bi₄Pd₃.**

2. In Fig. 1, it is noted that the data of TM are obtained from the maximum in $\chi_{\text{ai}}(T)$ (triangles) and B (stars) from Fig. 5. There is a maximum in χ_{ai} (B, T= 0.63 K) at around 11 T while there is no maximum in χ_{ai} (B, T=1.5 K). Therefore, I do not trust the star plotted in Fig. 1.

Response to (2): We believe the referee missed the maximum in χ_{ai} at 1.5 K because it is very close to the maximum in χ_{ai} at 0.63 K. In Fig. 5, we mark the maximum in χ_{ai} at 0.63 K with a purple line and the maximum in χ_{ai} at 1.5 K with a blue line. We are absolutely confident of the stars plotted at these temperatures in Fig. 1.

3. In the inset of Fig. 2a, the unit in the right-hand-axis is lacking. “Ce₃Bi₄Pd₃” on the top should be deleted because the data for Ce₃Bi₄Pt₃ are also plotted in this figure. In Fig. 2-6, the description of the scale such as 10⁻² (emu/mol.Ce) does not follow the standard style.

Response to (3): The lines have now been individually labeled. The dot has been removed between “mol” and “Ce”

4. The slope of the χ_{ai}^{-1} vs T data in the inset of Fig. 2b changes abruptly at around 250 K. The authors are requested to explain the reason.

Response to (4): The change in slope at 250K was found to originate from a glitch in the data, therefore it is not reliable. We have replaced the susceptibility-versus-temperature curve with one measured at 0.1 T. For the 5T data, we show only a truncated dataset up to 250 K.

5. In Fig. 3a, the thermal activation energy was estimated from a fit to the three data points of $n^*(T)$ at T = 7.5, 12.5, and 20 K. To obtain the activation energy in the Hall coefficient, I suggest the authors to present the Arrhenius plot of $\log R_{\text{H}}(T)$ vs 1/T.

Response to (5): Owing to the low residual carrier density ($n^* \sim 0.2\%$ per Ce), an Arrhenius plot cannot be made without first making an arbitrary subtraction. On the other hand, the

value of the gap we obtain from fitting is consistent with that we obtain from thermodynamic measurements (i.e. the heat capacity and also the susceptibility).

6. In Fig. 4, the $\rho(T)$ data in the fields of 12.5 -13 T become constant for $T < 6$ K. However, this field range is included in the FL state in Fig. 1.

Response to (6): In Fig. 1, the Fermi liquid regime is now shown only to extend as far down in field as the point where the resistivity transitions from metallic to insulating behavior (i.e. ~ 12.5 T).

7. The authors used the Schotte-Schotte formula to analyze the data of C/T under constant magnetic fields up to 9 T. It is necessary to show and analyze the data for $B > B_c = 11$ T to confirm the change in the density of states associated with the gap closure. For a Kondo semimetal, such analysis was reported by K. Izawa et al., PRB. 59, 2599 (1999).

Response to (7): The above manuscript of Izawa et al only contains data to 5T. Another older heat capacity manuscript [Izawa et al, J.Phys.Soc.Japan 65, 3119 (1996)] contains data to 14T, although this is still lower than $B_c=15$ T in CeNiSn, meaning that it does not demonstrate the effect of gap closure in the heat capacity.

We have not performed heat capacity measurements at fields higher than 9 T, but our work will stimulate the community to perform these measurements.

REVIEWERS' COMMENTS:

Reviewer #1 (Remarks to the Author):

I find the manuscript much improved over the first version and the authors have addressed all my questions. I recommend the manuscript for publication.

Reviewer #2 (Remarks to the Author):

After reading the new version of the manuscript, the comments and replies to all three referees, I think that the paper is improved and much clearer. In my opinion, the paper can be published. There are minor typing errors in the text.

Reviewer #3 (Remarks to the Author):

The authors answered to all my comments well. Now, the difference between this system with a Kondo semimetal is clear. I agree with the replies and recommend this paper to be published in Nature Communications.